# Distribution, Function, and Expression of the Apelinergic System in the Healthy and Diseased Mammalian Brain

**DOI:** 10.3390/genes13112172

**Published:** 2022-11-21

**Authors:** Martin N. Ivanov, Dimo S. Stoyanov, Stoyan P. Pavlov, Anton. B. Tonchev

**Affiliations:** 1Department of Anatomy and Cell Biology, Medical University-Varna, 9000 Varna, Bulgaria; 2Department of Stem Cell Biology, Research Institute, Medical University-Varna, 9000 Varna, Bulgaria

**Keywords:** APLNR, APJ, Apelin, ELABELA, CNS, CNS-associated diseases, neurogenesis, gene expression

## Abstract

Apelin, a peptide initially isolated from bovine stomach extract, is an endogenous ligand for the Apelin Receptor (APLNR). Subsequently, a second peptide, ELABELA, that can bind to the receptor has been identified. The Apelin receptor and its endogenous ligands are widely distributed in mammalian organs. A growing body of evidence suggests that this system participates in various signaling cascades that can regulate cell proliferation, blood pressure, fluid homeostasis, feeding behavior, and pituitary hormone release. Additional research has been done to elucidate the system’s potential role in neurogenesis, the pathophysiology of Glioblastoma multiforme, and the protective effects of apelin peptides on some neurological and psychiatric disorders-ischemic stroke, epilepsy, Parkinson’s, and Alzheimer’s disease. This review discusses the current knowledge on the apelinergic system’s involvement in brain physiology in health and disease.

## 1. Introduction

Apelin is a small, secreted protein first isolated from bovine stomach extracts and later identified as an endogenous ligand for the Apelin receptor [1]. It is secreted as a 77 aa precursor protein which is then cleaved into several mature forms that bind to the apelin receptor with different affinity and specificity.

Apelin receptors and their ligands are widely expressed in the mammalian brain [2,3,4,5,6,7]. Increasing interest in the apelinergic system has led to numerous discoveries and shed light on this system’s complex functional diversity. It is not only involved in the physiological processes in the brain, but it also plays a role in the pathophysiology of neuropsychiatric disorders. The system is a suitable pharmacological target: recent discoveries of novel synthetic Apelin-receptor ligands with increased half/life (LIT01-196, l-homoarginine aks l-hArg, and non-canonical amino acids l-cyclohexylalanine aka l-Cha, etc.) and antagonists have paved the way towards clinical trials [8,9,10]. For a comprehensive review of the newly synthesized APLNR ligands, refer to Fischer et al. [11].

## 2. Structure and Function of Apelin Receptor (APLNR; APJ)

Research on the apelinergic system history dates back to 1993 when the Apelin receptor (APLNR, also knowns as APJ) was first described and cloned from a human genomic library due to its significant structural similarity (~40% identity between amino acid sequences) to angiotensin II receptor (AT2R). However, because of its inability to bind Angiotensin II and because no endogenous ligand was known to interact with it at the time, APLNR was considered an orphan receptor [12]. This notion was updated in 1998 when a ligand named apelin was first isolated from bovine and human stomach extracts [1].

The human APLNR is a G protein-coupled receptor of class A (Rhodopsin-like receptors subclass A3) containing 380 amino acid residues with characteristics of the GCPR proteins family containing 7 TM α-helical segments [13,14]. Significant insight into its structure was recently described utilizing X-ray crystallography with a synthetic 17-residue mimetic [13]. The gene coding for APLNR is located on the long arm of chromosome 11 at locus 11q12.1.

APLNR expression has been described in other vertebrates, including rhesus macaque, mouse, rat, and cattle. In addition, lower vertebrates such as chicken, red-eared slider, zebrafish, and tilapia possess variants of APLNR receptors (APLNR1, 2, 2a, 2b, and 3a) with variable homology to the human receptor [3,6,15,16,17,18,19].

## 3. Structure and Function of Apelin Receptor Ligands

### 3.1. Apelin (APLN)

The first described endogenous ligand of the APLNR is apelin, generated from a 77 amino acids long inactive pre-proprotein named pre-proapelin. [1] It is encoded by a gene located in the X chromosome in rats (Xq35), mice (XA3.2), and humans (Xq25-26.1) [20]. The pre-proapelin gene contains three exons, with exons 1 and 2 being the coding regions [21].

The discovery of apelin was made by Tatemoto et al. in 1998 through extraction from bovine stomach tissue and was shown to act as a ligand for an orphaned at that time G-coupled protein receptor APLNR. The name Apelin comes from APJ Endogenous Ligand. Interestingly the homology of the protein is conserved across humans, mice, rats, and bovine [1,20].

Pre-proapelin has a secretory N-terminal signaling peptide (22 amino acid residues) and a C-terminal binding site. The last 55 amino acid residues in the binding peptide are highly conserved. Additionally, the many basic amino acid residues represent multiple cleaving sites for peptidases [1,22]. Pro-preapelin is subsequently cleaved, generating a range of peptides that bind to APLNR. However, each ligand can exert different tissue distribution, binding affinity, half-life, and functions (Figure 1A).

Upon cleavage, pre-proapelin is transformed into several fragments with sizes ranging from 55 to 13 residues (Figure 1A). The 55-residue is formed by removing the 22-amino acid N-Signaling peptide of the pre-proapelin. Apelin-55 is then further processed to generate shorter active isoforms like apelin-36, 17, and 13 through protease-mediated cleavage. Indeed, this is shown to be the case with converting proapelin to apelin-13 with the help of PCSK3 (FURIN) [23].

At first, the apelin ligand was isolated from bovine stomach extracts as a 36-amino-acid peptide capable of binding to CHO-cells expressing APLNR [1]. Additional experiments by the same group showed that shorter peptides (apelin-17, 13, and [Pyr^1^] apelin-13) exert stronger binding affinity to APLNR than apelin-36. Post-translational modifications of apelin-13 ([Pyr^1^] apelin-13) prevent ligand degradation by exopeptidases [22]. Forskolin-induced cAMP inhibition assays using CHO transfected with APLNR cDNA also showed that [Pyr^1^] apelin-13 prompted the most potent response. However, in another study, competition binding analysis in the same cell line (HEK293) established that apelin-13 has a higher affinity to the receptor than [Pyr^1^]apelin-13 [24]. Zhen et al. (2013) found that [Pyr^1^]apelin-13 is the most common ligand isoform in the blood [25]. For a long time, it was accepted that apelin-55 serves only as a precursor, but current evidence suggests that it may bind to APLNR with similar potency as apelin-17 and apelin-13 [26].

Several proteases (CD10 aka Neprilysin, angiotensin-converting enzyme 2 aka ACE2, and Kallikrein aka KLKB1) can regulate apelin by modification or inactivation [27,28,29]. The physiologically relevant action of metalloprotease CD10 is the cleavage of [Pyr^1^]apelin-13 between Arg4 and Leu5 (RPRL motif) and between Leu5 and Ser6, creating fragments 5–13 and 6–13, respectively [30]. Cleavage of the RPRL motif inactivates the protein and prevents it from binding to APLNR [27].

Angiotensin-converting enzyme 2 (ACE2) is a zinc-metalloprotease with carboxypeptidase activity that removes the C-terminal phenylalanine and produces apelin isoforms with modified action [28]. Cleaving of apelin-13 and apelin-36 by ACE2 produces two fragments, apelin-13_(1–12)_ or apelin-36_(1–35)_. Apelin-13_(1–12)_ binds to the APLNR and activates downstream pathways [31]. It is worth mentioning that ACE2 was recently identified as the main SARS-CoV-2 receptors. Saravi and Beer proposed that apelin peptides can be used as a potential drug for improving the outcome of COVID-19 lung and cardio-vascular injuries [32]. Apelin-17 was shown to be indirectly related to the COVID-19 severity [33].

Kallikrein is a serine protease that cleaves apelin-17 between Arg3 and Arg4 into two isoforms, apelin-17_(1–3)_ and apelin-17_(4–17)_ fragments. Those fragments bind to the receptor with high affinity but cannot activate the Ca^2+^ ions mobilization [29].

### 3.2. Apela/ELA (Apelin Receptor Early Endogenous Ligand/Elabela/Toddler)

The human ELABELA gene is located on chromosome 4 and contains three exons. The transcript, previously believed to be a non-coding RNA, contains an open reading frame (ORF) and encodes a pre-proprotein with a length of 54 amino acids, containing a 22 amino acids secretory signaling terminus and a 31 amino acids mature portion. This protein is highly conserved across the species with nearly perfect homology of the C-terminus’s last 13 amino acid residues [34,35,36].

ELABELA (Toddler) was discovered and described as an essential regulator of heart development [34,35]. In embryos, ELABELA acts as an early developmental signal required for the migration of mesendodermal cells. ELABELA^-/-^ knock-out mice show cardiac agenesia or form only a rudimentary heart. This phenotype resembles the effects observed in APLNR knock-out mice [34,35]. Apelin and ELABELA have sequence similarity of 25%, isoelectric points above 12, and are rich in basic amino acid residues. These observations led to the experimental evidence and conclusion that ELABELA can also activate the APLNR [34,35,37].

Additional experiments utilizing HEK293 cells transfected with lentivirus containing *APLNR* enchased by GFP and treated with ELABELA show that after binding to the APLNR, ELABELA causes transient internalization of the receptor. Similarly, the binding of ELABELA to APLNR-expressing cells acts on Gi-coupled proteins leading to stimulation of ERK 1/2 phosphorylation, an increase in calcium mobilization, and a decrease in cAMP production [38,39]. Interestingly, competition binding analysis showed that ELABELA binds with a higher affinity to APLNR than apelin [39].

Chng et al. have speculated on the cleavage sites of the ELABELA peptide because of the presence of dibasic amino acids: Arg9–Arg10 and Arg20–Arg21 [34]. Indeed, incubation of ELA-32 in the presence of PCSK3 (FURIN) leads to the generation of fragment ELA-11_(22–32)_ [40] (Figure 1B). In addition, incubation of ELABELA in rat plasma induces the formation of two other bioactive fragments- ELA_(1–9)_ and ELA-22_(11–32)_ with a cleavage site Arg9-Arg10 and Arg10-Lys11 [41].

Some of those smaller fragments can bind effectively to the receptor and activate the Gαi1 pathway, recruiting β-arrestins, leading to the internalization of the receptor with efficacy similar to that of ELABELA (parent form) and Apelin-13 [40,41]. ELA-11 is a notable exception, with reduced receptor binding affinity and β-arrestin recruitment but with the capability to inhibit cAMP production [42]. The plasma half-life of ELA is very short (t_1/2_ = 2 min), similar to Apelin [41]. The plasma concentration of apelin-13 is in the range of 0.13 ± 0.05 ng/mL [43].

## 4. Transcriptional Regulation of *APLNR* and *Apelin*

Multiple transcriptional factors are involved in the regulation of the apelin receptor gene expression, including Sp1 (stimulating protein-1), glucocorticoid, estrogen receptors, and CCAAT enhancer-binding protein (C/EBP) [44]. Additionally, insulin can also affect the expression of both apelin and APLNR [45]. Upon knock-out of *Apelin*, the expression of APLNR was reduced, suggesting that *Apelin* regulates the expression of APLNR [46].

Based on the data so far, transcriptional factors regulating apelin gene expression are upstream transcription factor 1/ upstream transcription factor 2 (USF1/USF2), signal transducer and activator of transcription 3 (STAT3), hypoxia-inducible factor 1-α (HIF1-α), TNF-α (tumor necrosis factor-α) collaborative binding interaction between RARα (retinoic acid receptor α), KLF5 (*Krüppel*-*like* factor 5) and Sp1 (stimulating protein-1) [47,48,49,50,51]. Insulin is also shown to increase apelin expression. [45] Down-regulation of apelin is associated with the transcriptional factors ATF4 and its binding partner C/EBP-b via the p38 MAPK pro-apoptotic signal pathway [52].

Moreover, various substances such as resveratrol metabolites, andrographolide, and insulin can increase the expression of mRNA of *Apelin*, while the application of corticosteroids can downregulate it [53,54,55].

## 5. Signaling Pathways Associated with Activation of APLNR

APLNR is coupled with heterogenous guanine nucleotide-binding proteins (Gαi/o, Gαq/11, Gαs, and Gα12/13), which upon activation of the receptor, can regulate diverse intracellular events. (Figure 2) It is possible that the APLNR is also coupled with inhibitory G-coupled protein (Gi) [1].

APLNR is associated with different G-coupled proteins Gαi/o, Gαq/11, Gαs, and Gα12/13. Activation of the receptor and recruitment of Gαs activates Adenylyl Cyclase leading to the production of PKA. Adenylyl cyclase can be inhibited by Gαi/o. Gαi/o and, on the other hand, is responsible for downstream reaction associated with cell cycle progression, inhibition of autophagy, and cell survival and response to injury through activation of PI3K/Akt/mTOR pathways. mTOR can phosphorylate P70S6K kinase, leading to cell migration and proliferation. Additionally, mTOR can be phosphorylated (Ser2448), leading to apoptosis and autophagy inhibition. Gαi/o can also activate MEK1/2/ERK1/2 and phospholipase Cβ leading to an increase in cell survival due to inhibition of apoptosis. In addition, Gαq/11 and Gq also can activate phospholipase Cβ. Gα12/13 mediates cytoskeleton remodeling by activating RhoGEF/Rhoa pathway. Apelin is also involved in inducing self-renewal in cancer stem cells. A possible mechanism involving the apelinergic system is associated with increased gene suppression and enhanced stem cell self-renewal. GSK3b increases the stability of KDM1A via phosphorylation at s683, which can interact with USP22. KDM1A is responsible for the demethylation of histone H3K4 downregulating genes (BMP2, CDKN1A, and GATA6) associated with stem cell self-renewal and tumorigenesis. When bound to APLNR, Apelin-13 recruits GRK2 and β-arrestins leading to internalization of the receptor via transferrin and rapid recycling back to the cell membrane. Apelin-36, on the other hand, also recruits GRK2 and β-arrestins, but in this case, the APLNR is targeted for lysosomal degradation. Apelin is also callable of attenuating the mitochondrion permeabilization caused by NMDAR activation, decreasing the generation of ROS, Cytochrome C, and Caspase-3, thus, inhibiting apoptosis. Another attenuation mechanism is the Ca2^+^-dependent Casein kinase-2 (CK2) phosphorylation of NR2B subunit at S1480, leading to decreased activity of NMDAR. Additionally, the administration of apelin-13 can inhibit the effect of the GRP78/CHOP pathway and caspase-12 cascade associated with ER stress. Abbreviations: NMDAr, N-methyl-D-aspartate receptor; NR2B, The N-methyl-D-aspartate receptor subunit 2B; CK2, Casein kinase 2; cAMP- cyclic Adenosine monophosphate; ATP, Adenosine triphosphate; PKA, protein kinase A; PI3K, Phosphoinositide 3-kinase; Akt, Protein kinase B; mTOR, mammalian target of rapamycin; MEK1/2, Mitogen-activated protein kinase kinase 1/2; ERK 1/2, extracellular signal-regulated kinase; RhoGEF, Guanine nucleotide exchange factor for Rho/Rac/Cdc42-like GTPases; Rhoa, Ras homolog family member A; PLC β2, Phospholipase C β2; IP3, Inositol trisphosphate, PIP2, Phosphatidylinositol 4,5-bisphosphate; DAG, diacylglycerol; PKC, Protein kinase C; GRK2, G Protein-Coupled Receptor Kinase 2; BAX, Bcl-2-associated X protein; BID, BH3 interacting-domain death agonist; BIK, Bcl-2-interacting killer; ROS, Reactive oxygen species; CHOP, C/EBP Homologous Protein; GRP78, Glucose regulated protein-78; GSK3β, Glycogen synthase kinase-3 β, KDM1A, Lysine-specific histone demethylase 1A; USP22, Ubiquitin Specific Peptidase 22.

Those G-proteins are associated with distinct cellular events like activation or inhibition of adenylate cyclase by Gαs or Gαi/o, respectively, increase in intracellular Ca^2+^ with activation of phospholipase Cβ by Gαq, actin cytoskeleton remodeling by Rho activated by Gα12/13 and others. Additional pathways involving EGFR and -β-arrestin signaling are also relevant to downstream reactions associated with APLNR activation.

Coupling to Gαi/o inactivates the Adenylyl Cyclase-reducing protein kinase A (PKA) [2,5,22,24,56]. APLNR/Apelin interactions are shown to enhance downstream effects related to the kinases Akt and ERK1/2, which are functionally related to cell survival and injury protection [57]. Apelin-13 enhances the phosphorylation of Akt while reducing phosphorylated ERK1/2, demonstrating a potent neuroprotective effect and promoting cell survival in a serum deprivation (SD) neuronal model of apoptosis in cultured cortical neurons. [58] Interestingly, upon exposure to apelin ligands, neurons demonstrate an increase in intracellular calcium [59]. On the other hand, Gαs can activate Adenylyl Cyclase, which increases cAMP, inducing PKA activation [60,61].

Activation of Gα12/13 leads to activation of Phospholipase C, which hydrolyses phosphatidylinositol 4,5-bisphosphate (PIP_2_) into diacylglycerol (DAG) and inositol 1,4,5-trisphosphate (IP_3_). Increased Ca^2+^ concentration, a result of IP_3_ binding to the endoplasmic reticulum, is required in addition to DAG to activate PKA [60,61]. Apelin can also regulate cell cycle progression by activating p70S6K through PI3K/Akt/mTOR pathway [62]. mTOR can be phosphorylated through PI3K/Akt at the Ser2448 position leading to the inhibition of autophagy and caspase-3 activation associated with apoptosis [63].

Different ligands exhibit different overall effects on downstream processes like receptor phosphorylation, recruitment of GRK2, and β-arrestins [60,64].

Of note, when internalized, both apelin-13 and 36 colocalize with transferrin. Interestingly, the binding of apelin-36 to the receptor recruits β-arrestin-1, which leads to the internalization of the receptor via early Rab4 endosomes. On the contrary, the apelin-13 binding leads to transient internalization of the receptor followed by rapid recycling to the cell surface. The internalization process is not associated with β-arrestin complex formation, as observed during apelin-36 internalization [65,66]. This suggests ligand-depending trafficking mechanisms and downstream effects of the system.

Studies have shown that activating pro-survival pathways (ERK1/2 and AKT) by pre-treating hippocampal cultures with apelin ligands rescues the neurons from N-methyl-D-aspartic acid (NMDAr) receptor-mediated excitotoxicity injury [57]. Mitochondrial depolarization and increased permeabilization can increase ROS and cytochrome C, which are the main starting points for both apoptosis and necrosis. Interestingly, infusion of apelin-13 in serum-deprived cortical neuron cultures can attenuate ROS generation by stabilizing the mitochondrial membrane [58]. The underlying mechanism of this process involves IP3, PKC, MEK1/2, and ERK1/2 Signaling pathways.

Since NMDA receptors include glycine binding subunits-NR1 and glutamate binding subunits-NR2 (NR2A–NR2D), a potential Casein kinase-2 (CK2) or Ca^2+^-dependent phosphorylation of NR2B S1480 by the APLNR pathway provides additional mechanisms of protection against excitotoxicity [67]. Moreover, apelin ligands inhibit HIV-associated neurotoxicity and apoptosis in neurons [57].

## 6. APLNR, Apelin, and ELABELA Expression in the Normal Brain

Brain distribution of APLNR and its endogenous ligands has been extensively studied utilizing different molecular and histo-anatomical techniques. APLNR and Apelin are present in the mammalian central and peripheral nervous systems. The topographical localization of APLNR and Apelin in the brain suggests multiple roles of this system in neurogenesis, pituitary hormone release, body fluid homeostasis, regulation of blood pressure, feeding behaviors, etc. [5,16,56,68]. Regionally, APLNR is expressed in restricted areas of the cerebral cortex like frontal, temporal, occipital, piriform, and entorhinal cortices (Figure 3, Table 1) [2,5,12,20,57,68,69,70]. Compared to APLNR, the overall levels of *Apelin* mRNA measured with RT-PCR in the cerebrum is higher [70].

Using a variety of molecular biology techniques (qPCR, RT-PCR, WB, NB) for detecting APLNR and Apelin, multiple studies have shown that both are present in the dentate gyrus and the hippocampus proper (Cornu Ammonis, CA), Apelin having a higher expression level as compared to the APLNR [2,5,12,19,20,57,68,69,70]. The subgranular zone of the dentate gyrus is one of the main niches for postnatal neurogenesis, and it will be interesting to investigate if the APLNR/Apelinsystem can modulate its cellular output. Our recent data in primates show that APLNR is expressed in the other major neurogenic niche: the anterior subventricular zone along the cerebral lateral ventricle (SVZa) [16]. Probably, the cells expressing APLNR in SVZa represent the neural progenitor subpopulation [16]. Interestingly, APLNR shows a strong expression in the caudate nucleus [16], another region of adult neurogenesis in primates and humans [71]. APLNR is expressed in the thalamus [2,69]. In the hypothalamus, both apelin and APLNR have been localized [2,3,5,20,70]. Their expression was restricted to the supraoptic (SON) and the paraventricular nucleus, both magnocellular and parvocellular parts, contributing to the maintenance of fluid homeostasis [3,5,6,20,68,72]. The apelinergic system has also been detected in other brain areas, including the substantia nigra, cerebellum, preoptic area, pituitary gland, medulla oblongata, pons, and the spinal cord [2,5,19,24,69,70,72,73]. At the cellular level, APLNR has been detected in neurons, oligodendrocytes, and astrocytes but not in microglia, while Apelin has been detected in neurons but not astrocytes and microglia [2,57,58,59,74].

**Table 1 genes-13-02172-t001:** Table with different brain regions in the human brain with known expression of APLNR and Apelin ligand.

	Human		
Tissue	Apelin (Preproapelin)	APLNR	References
Whole Brain	Unknown	^+^	[7]
Frontal Cortex	Unknown	^+^	[2,69]
Temporal Cortex	^+^	^+^	[2,75]
Striatum (Overall)	Unknown	^+^	[69]
Putamen	^+^	^+^	[2]
Caudate nucleus	^+^	^+^	[2]
Accumbent nucleus	^+^	^+^	[2]
Corpus callosum	^+^	^+^	[2,69]
Hippocampus (GD + CA)	^+^	^+^	[2,69,75]
Amygdala	^+^	^+^	[2,69]
Thalamus	^+^	^+^	[2,69]
Hypothalamus (Overall)	^+^	^+^	[2]
Substantia nigra	^+^	^+^	[2,69]
Cerebellum	^+^	^+^	[2]
Medulla oblongata	Unknown	^+^	[69]
Spinal cord	^+^	^+^	[2,69]

## 7. A Role of the Apelinergic System in Brain Diseases

Mounting evidence suggests that the apelinergic system is a prominent player in the pathogenesis of different neuronal and mental diseases, such as stroke, epilepsy, Alzheimer’s disease, and Parkinson’s, among others. 

### 7.1. Apelinergic System Involvement in Ischemic Stroke

Ischemic stroke is the most common cause of disability and death worldwide [76]. Damage caused by cerebral blood vessel occlusion leads to the regional increase in Ca^2+^ (via NMDAR activation), depolarization of the mitochondrial membrane, caspase activation, neuronal cell death, and cerebral edema. Infusion of apelin-13 in mice reduces the infarct zone volume [77], cerebral edema, and caspase-3 activation but does not alter the neurological deficits [78]. Apelin-36 in lower concentrations can also reduce the infarct volume, but unlike apelin-13, it also improves neurological function after ischemia/reperfusion injury. LY294002, a potent inhibitor of PI3K, reduced the phosphorylation of Akt, thus, lowering the activity of the PI3K/Akt pathway activated by the APLNR ligands. Applying this substance to the ischemic stroke model treated with apelin-13 or 36 elevates the pro-apoptotic proteins caspase-3 and BAX, confirming that the antiapoptotic effect of apelin-36 is induced by PI3K/Akt pathway [79].

Apelin-13 treatment significantly reduced the levels of neutrophil infiltration in the ischemic penumbra and the levels of the pro-inflammatory mediators IL-1β, TNF-α, and ICAM-1. Moreover, it can also lower the number of cells activated in the penumbral region, thus, inducing a neuroprotective effect by blocking or suppressing neuroinflammation [77,80]. Intranasal administration of Apelin-13 effectively reduced the number of apoptotic cells and of activated microglial cells, increasing the expression of antiapoptotic factors (Bcl-2). It could also reduce the pro-inflammatory cytokines and chemokines TNF-a, IL-1b, MIP-1a, and MCP-1 and increase the anti-inflammatory cytokine IL-10. Angiogenesis in the peri-infarct region can be explained by the enhanced activity of pro-angiogenic factors VEGF and MMP9, which were also elevated after treatment with apelin-13. Because of the enhanced angiogenesis after treatment, better recovery was reported compared to non-treated animals [81]. Upon treatment with apelin, an upregulation of the expression of VEGF and VEGF-2 can be observed. This elevation is associated with the protective effects of apelin, mediated by ERK and PI3K/Akt pathways, which can be blocked by intraventricular injection with an anti-VEGF antibody [80].

Following cerebral ischemia in primates, *APLNR* and *Apelin mRNA* was strongly induced in monkey SVZa and caudate nucleus [16].

### 7.2. Apelinergic System Involvement in Epilepsy 

Neurons in the mammalian neocortex are either excitatory, glutamatergic projecting neurons or inhibitory, GABAergic interneurons that branch in the local circuits. A disbalance in the excitation levels leads to pathological hyperexcitability manifested by spontaneous and recurrent seizures [82,83].

Extended epileptic periods and poorly managed or drug-resistant epilepsy can cause neuronal loss either by apoptosis or necrosis. The observed overexpression of Apelin in patients with drug-resistant temporal lobe epilepsy and rats with lithium–pilocarpine-induced epilepsy may be a compensatory mechanism [75]. Apelin can salvage the hippocampal neurons from the effects of excitotoxicity by downregulating metabotropic Glutamate Receptor-1 (mGluR1), increasing phosphorylation of Akt, and upregulating Bcl2, thus, reducing caspase-3 activation [84]. Treatment with brain-specific micro-RNA-182 (miR-182) that blocks Apelin leads to increased apoptosis in epilepsy models. Blocking miR-182 can increase the effects on Apelin, lower pro-apoptotic proteins (Bax; caspase-3), and increase the antiapoptotic ones (Bcl-2) [84].

Treatment with apelin-13 in an experimental rat epilepsy model prevented the induction of seizures and neuronal loss. This effect is lost when F13A, an APLNR receptor antagonist, is applied [85]. Apelin can exert a level of neuroprotection in the PTZ model of epilepsy thanks to its ability to maintain mitochondrial potentials, reduce intracellular Ca^2+^, and inhibit ROS generation and COX2 (Cyclooxygenase 2) [86].

### 7.3. Apelinergic System Involvement in Neurogenesis and Glioblastoma Multiforme

Glioblastomas are brain tumors showing high invasiveness, angiogenesis, and an unusual tumor environment. There is substantial evidence showing that Glioblastoma multiforme is derived from SVZa stem cells [87].

Apelin is secreted from the endothelial cell near Glioblastoma stem-like cells (GSCs). It mediates self-renewal, but it is not associated with proliferation. Apelin protein expression is also correlated with the levels of vascularization of GBM [88].

Silencing the apelin Signaling pathway either by knocking down or blocking the APLNR reduces tumor volume, vascularization, and proliferation [89]. GSC are in a quiescent state maintained by the vascular niche in the tumor, which is the main reason for the inefficiency of chemotherapies [90]. Interestingly, applying an antagonist of APLNR in combination with chemotherapies improves the response and decreases the GSC numbers. This effect is possibly mediated by activation of GSK3β (Glycogen synthase kinase-3 pathway) [88,89,91]. Nuclear GSK3β phosphorylates KDM1A at s683, which can interact with USP22, thus, increasing the stability of KDM1A. KDM1A is responsible for the demethylation of histone H3K4 leading to the downregulation of genes (*BMP2*, *CDKN1A*, and *GATA6*) associated with stem cell self-renewal [92].

ELABELA was also shown to be expressed in GSCs. Moreover, brain tumor datasets have shown that expression levels of ELABELA are linked to tumor grading and patient survival [93].

Current therapies relying on anti-VEGF mAb usually target tumor angiogenesis. Unfortunately, such therapies have not increased patient survival [94]. These treatments have been shown to decrease the apelin expression inside the tumor, thus, increasing its invasiveness [95]. Interestingly, using a partial agonist for APLNR (apelin-F13A) combined with anti-VEGF therapy lessens the invasiveness and angiogenesis properties of GBM [95,96].

### 7.4. Apelinergic System Involvement in Alzheimer’s Disease (AD)

Alzheimer’s disease is a progressive neurodegenerative disorder characterized by the deposition of intracellular senile plaques composed of insoluble neurofibrillary tangles and extracellular amyloid β (Aβ) peptides. Neuronal loss in the hippocampus and neocortex leads to memory loss and cognitive impairments [97,98].

In newly discovered AD patients, the levels of Apelin-13 were lower compared to healthy individuals [43].

Apelin-13 can reduce memory deficits in a mouse model of Alzheimer’s disease. [63,99] Aβ deposition in neurons induces apoptosis and autophagy, which can be attenuated by Apelin-13 treatment. The molecular basis of these neuroprotective effects in AD models is: (i). Decreased autophagy pathway (e.g., LC3II/I), (ii). Increase of autophagic clearance (HDAC6), (iii). Decreased apoptosis (caspase-3), and (iv). Increasing survival of neurons through the mTOR pathway [63].

Neuroinflammation plays a critical role in the pathophysiology of Alzheimer’s disease. Important components of the neuroinflammation response, including microglial and astroglial activation and pro-inflammatory cytokine (e.g., IL-1β and TNF-α) production are attenuated from Apelin-13 [99].

Apelin can also increase the expression of hippocampal neurotrophins/neurotrophin receptors, such as Brain-Derived Neurotrophic Factor (BDNF) and Tropomyosin receptor kinase B (TrkB), which are typically at low levels in Alzheimer’s mouse models. Blocking the TrkB receptor with an apelin antagonist, K252a, blocked the apelin-13 effects, showing that the beneficial effects of apelin in the hippocampus are mediated by activation of the BDNF/TrkB Signaling pathway. Synaptophysin (SYP) generaly used for evaluating synaptic transmission plasticity is downregulated in AD and restores its normal levels upon reapplication of apelin-13 [99]. Tissue necrosis is also initiated in AD by activation of the proteins RIP1 and RIP3, controlled by TNF-α. Reduction of RIP1, RIP3, and TNF-α is observed when apelin is applied [100]. Wan et al. have provided an in-depth review of the role of apelin in AD and its mechanism of neuroprotection [101].

### 7.5. Apelinergic System Involvement in Parkinson’s Disease (PD)

Parkinson’s disease (PD) is a neurodegenerative disorder affecting the dopaminergic neurons in the substantia nigra. It manifests with motor dysfunctions, including muscle rigidity, tremor, slow movement, and cognitive impairments, including depression, anxiety, and in later stages, dementia. The main histological hallmark of the disease is the aggregation of a misfolded protein called α-synuclein, which accumulates and becomes cytotoxic [102,103]. Additionally, factors such as mitochondrial dysfunction, inflammation, oxidative stress, and synaptic dysfunction can play a crucial role in the pathophysiology of the disease. To show the role of apelin-13, Pouresmaeili-Babaki et al. used SH-SY5Y cells treated with 6-hydroxydopamine (6-OHDA), which is a widely used cell model for PD. Upon treatment with 6-OHDA, dopaminergic cell death can be observed. Application of Apelin-13 is capable of inhibiting cytochrome-3 release and activation of caspase-3, effects through activation of APLNR/PI3K/Akt Signaling pathway [104]. The same group was able to show also that Apelin can improve memory and cognitive deficits in a Parkinson’s disease model treated with 6-hydroxydopamine (6-OHDA) [105].

Another study, utilizing the same SH-SY5Y cell line but induced cell damage by applications of 1-methyl-4-phenyl-pyridine (MPP+) showed that apelin-13 could attenuate the neurotoxicity and the Endoplasmic Reticulum Stress (ER stress), the level of GRP78, CHOP and cleaved caspase-12 and significantly increase the levels of phosphorylated ERK1/2, thus, preventing the apoptosis [106]. Similarly, another study using 1-methyl-4-phenyl-1,2,3,6-tetrahydropyridin (MPTP) to induce Parkinson-like damage has shown that apelin-13 significantly increases autophagy by upregulation of LC3B and Beclin1 and down-regulation of p62. Apelin-13 was also capable of inhibiting the effect of the GRP78/ IRE1α/XBP1s/CHOP pathway associated with ER stress [107].

The neuroprotective effect of apelin was also shown in the methamphetamine PC12 cell model. Applying methamphetamine increased the generation of ROS, autophagy, and apoptosis, which were reduced by apelin [108]. Furthermore, some evidence suggests that it can also alleviate motor deficits [107] and prevent pathological alterations to the synaptic elements in the striatum and substantia nigra [109].

## 8. Conclusions and Future Directions

The components of the apelinergic system are widely expressed in the adult brain, and interaction between APLNR and Apelin isoforms or ELABELA are responsible for numerous physiological functions. Apelin and ELABELA are modified and processed into multiple functional and non-functional isoforms. Although the homology between apelin isoforms is similar, and they can all activate APLNR, the activated downstream signaling cascades may vary. The studies we discussed in this review show the role of the apelinergic system in the physiology of the brain and the pathophysiology of brain-related diseases. In animal models, the apelinergic system can activate critical signaling pathways related to cell survival, response to injury, reduction of apoptosis, cell cycle regulation, and stem cell biology.

Future work will need to address the system’s function in relation to gene regulation, stem cell biology, neuronal development, and the related implications in those areas.

The expression of Apelin and APLNR in the hippocampus and the SVZ along the lateral wall of the cerebral lateral ventricle suggests that the apelinergic system may affect neural regeneration.

Finally, the identification and characterization of novel analogs and ligands with an increased half-life, specificity, and binding strength will advance the quest for novel therapeutic approaches in treating neuropsychiatric disorders and will increase our understanding of how the components of the system interact.

## Figures and Tables

**Figure 1 genes-13-02172-f001:**
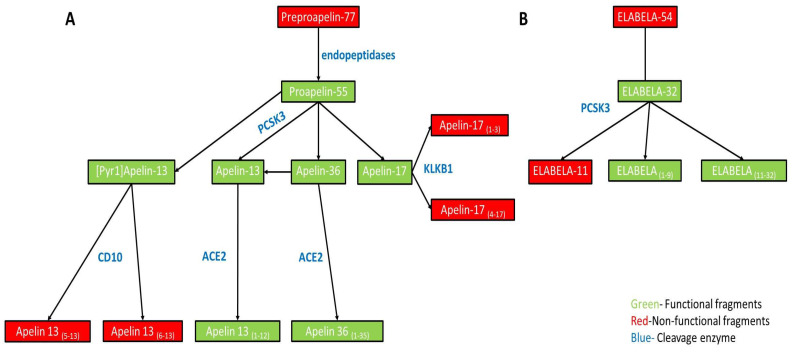
Enzymatic processing of Apelin/ELABELA. (**A**). The inactive pre-proapelin (77aa) is processed via endogenous endonucleases to Apelin-55, which can bind to APLNR. Apelin-55 can be additionally processed, generating four isoforms: Apelin-13, Apelin-17, Apelin-36, and [Pyr^1^] apelin-13. The generation of Apelin-13 is achieved by PCSK3 (FURIN). All four isoforms are capable of binding to the APLNR. CD10 (Neprilysin) is capable of inactivating [Pyr^1^] apelin-13, creating two inactive Apelin-13 isoforms (5–13aa and 6–13aa). On the other hand, Angiotensin-converting enzyme 2 (ACE2) converts Apelin-13 and Apelin-36 to active forms, Apelin-13_(1–12)_ and Apelin-32_(1–35)_. (**B**). ELABELA gene codes for a non-functional 54aa-long pro-protein, which generates an active 32aa-long protein upon processing. ELABELA-32 can generate three fragments, ELABELA-11, an inactive form, generated with the help of PCSK3, and two functional ones, ELABELA_(1–9)_ and _(11–32)_, with the activity of unknown proteases. Abbreviations: PCSK3, proprotein convertases subtilisin/kexin type; CD10, Neprilysin; ACE2, Angiotensin Converting enzyme-2; KLKB1, Kallikrein.

**Figure 2 genes-13-02172-f002:**
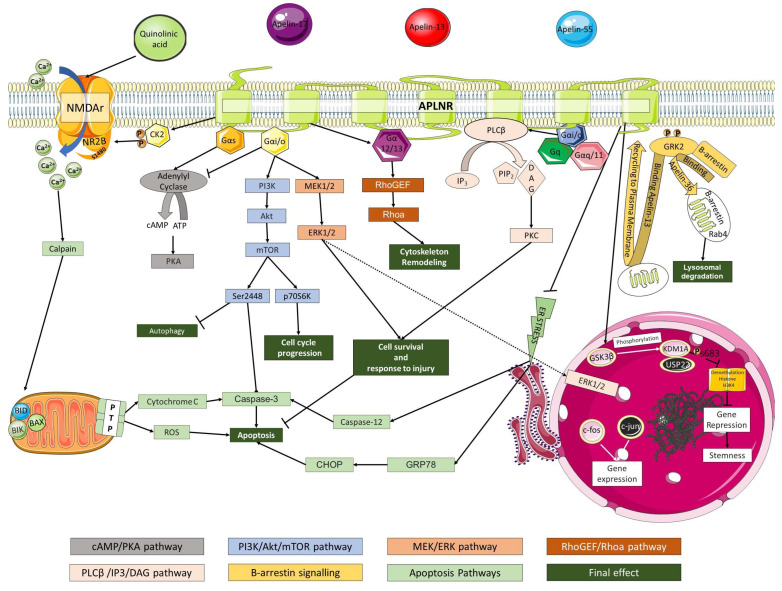
Signaling pathways associated with activation of APLNR.

**Figure 3 genes-13-02172-f003:**
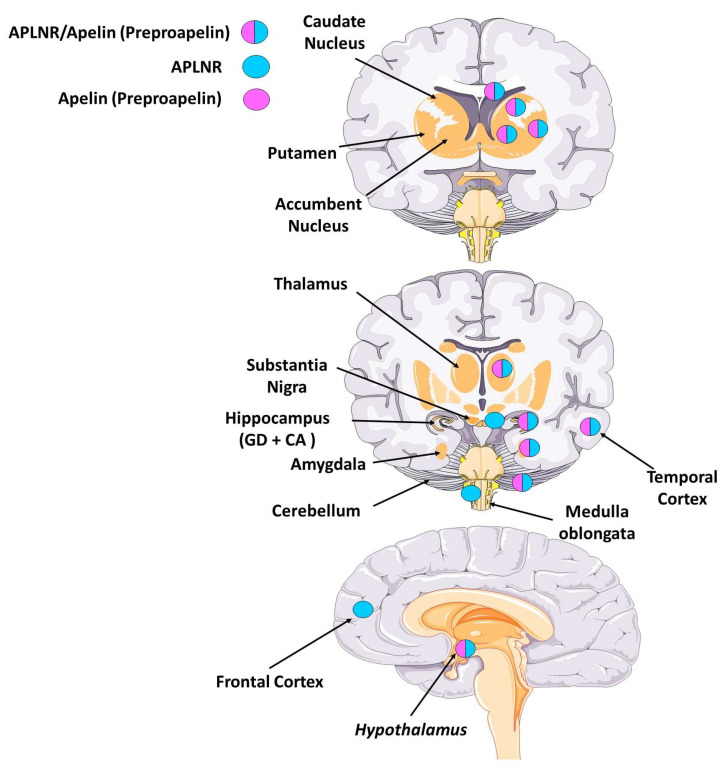
**Expression of APLNR and Apelin ligand in the normal human brain.** Schematic representation depicting the adult human brain at different levels showing the structures in which APLNR and Apelin ligand are expressed.

## Data Availability

Not applicable.

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
