# Peer review of "Distribution, Function, and Expression of the Apelinergic System in the Healthy and Diseased Mammalian Brain"

_genes, 2022, doi:10.3390/genes13112172_

Round 1

Reviewer 1 Report

Very well-described manuscript for further implication.

Author Response

 Point 1: Very well-described manuscript for further implication.

Responses 1:  Dear reviewer, thank you for this kind comment regarding the present manuscript.

Reviewer 2 Report

The review submitted by Ivanov and coll. is well articulated and covers the proposed field as summarized by the title.

It mainly suffers from English, which must be revised, especially in the second part of the review, dedicated to the pathophysiological role of Apelin/APJ.

Just for example, see at pag. 10, lines 343-345 or lines 347-349, in which some parts of the sentences are incomplete. Again, on pag. 10, lines 372-374, "decrease COX2, am enzyme... was also reduced ..." is not understandable. A similar error can be found on page 11, lines 428-430. In  line 434 "neuroprotective" alone is inappropriate.

Other suggestions:

-On page 1, line 26, instead of "leaving" it could be better "releasing"

-On page 1, when the Authors indicate the name of the receptor for Apelin should point out that there are two names, namely APJ and APLRN, as they first use APLRN and in the third chapter they start to use APJ  without further explanation.

- On pag 4, lines 129-135, che concept that ELABELA influences mesodermal cell migration is abundantly repeated and this should be simplified, considering that the references cited are the same  [34, 35].

Author Response

Point 1: The review submitted by Ivanov and coll. is well articulated and covers the proposed field as summarized by the title.

Response 1: Thank you for this statement about the construction of our manuscript.

Point 2: It mainly suffers from English, which must be revised, especially in the second part of the review, dedicated to the pathophysiological role of Apelin/APJ.

Just for example, see at pag. 10, lines 343-345 or lines 347-349, in which some parts of the sentences are incomplete. Again, on pag. 10, lines 372-374, "decrease COX2, am enzyme... was also reduced ..." is not understandable. A similar error can be found on page 11, lines 428-430. In line 434 "neuroprotective" alone is inappropriate.

Response 2: Thank you for your recommendations! We will correct and revise the manuscript to your suggestions.

Point 3: On page 1, line 26, instead of "leaving" it could be better "releasing".

Response 3: Thank you for this correction suggestion; the change has been implemented.

Point 4: On page 1, when the Authors indicate the name of the receptor for Apelin should point out that there are two names, namely APJ and APLRN, as they first use APLRN and in the third chapter they start to use APJ without further explanation.

Response 4: Thank you for this correction suggestion. We have added the aliases for both APLNR and Apelin. For the purposes of this review, we have chosen to use only the names APLNR and Apelin exclusively in the text.

Point 5: On page 4, lines 129-135, the concept that ELABELA influences mesodermal cell migration is abundantly repeated and this should be simplified, considering that the references cited are the same [34, 35].

Response 5: Thank you for this correction suggestion. We have simplified the mentioned part of the manuscript.